# Research on Quality Characterization Method of Micro-Injection Products Based on Cavity Pressure

**DOI:** 10.3390/polym13162755

**Published:** 2021-08-17

**Authors:** Quan Wang, Xiaomei Zhao, Jianpeng Zhang, Ping Zhang, Xinwei Wang, Chaofeng Yang, Jinrong Wang, Zhenghuan Wu

**Affiliations:** 1National Local Joint Engineering Laboratory of Intelligent Manufacturing Oriented Automobile Die & Mold, Tianjin University of Technology and Education, Tianjin 300222, China; zxm01292119@163.com (X.Z.); zzzzjpppp@163.com (J.Z.); wangxinwei96@163.com (X.W.); 13821532033@163.com (C.Y.); wang451330609@163.com (J.W.); wzh46817@163.com (Z.W.); 2School of Automobile and Transportation Engineering, Guangdong Polytechnic Normal University, Guangzhou 510665, China; cathyzp2002@163.com

**Keywords:** micro-injection molding, cavity pressure, switchover, weight, peak cavity pressure, area under curve of cavity pressure

## Abstract

The cavity pressure in the injection molding process is closely related to the quality of the molded products, and is used for process monitoring and control, to upgrade the quality of the molded products. The experimental platform was built to carry out the cavity pressure experiment with a micro spline injection mold in the paper. The process parameters were changed, such as V/P switchover, mold temperature, melt temperature, packing pressure, and injection rate, in order to analyze the influence of the process parameters on the product weight. The peak cavity pressure and area under the pressure curve were the two attributes utilized in investigating the correlation between cavity pressure and part weight. The experimental results show that the later switchover allowed the injection to proceed longer and produce a heavier tensile specimen. By comparing different cavity pressure curves, the general shapes of the curves were able to indicate different types of shortage produced. When the V/P switchover position is 10 mm, the coefficient of determination (R^2^ value) of part weight, for the peak cavity pressure and area under the curve, were 0.7706 and 0.8565, respectively. This showed that the area under the curve appeared to be a better process and quality indicator than the peak cavity pressure.

## 1. Introduction

Injection molding is an important method in plastic molding [1,2,3]. With the development of the economy and the improvement of living standards, people have higher and higher requirements for the quality of products, and the technology of injection molding machines is gradually developing towards the direction of intelligence and precision [4,5,6].

Injection molding is a cyclic process, consisting of the following four phases: filling, melt compressing (or packing), holding, and cooling, as shown by the typical cavity pressure profile in Figure 1. The filling process starts at point A. The cavity pressure signals begin at point B—where the melt plastics touch the pressure sensor for the first time—and then the pressure increases steadily as the filling proceeds. The filling phase is complete at point C, where the cavity is only volumetrically filled by the melt, without being compressed. The packing process then embarks and the pressure rises rapidly to the peak value (Pmax) at point D. Thereafter, the melt within the cavity is maintained at an assigned pressure during the holding phase, when additional plastic melt can be packed into the cavity, to compensate for the plastic shrinkage caused by cooling, so as to have the mold completely filled. This process continues until the gate is frozen, as marked at point E. The final cooling phase comes afterwards and continues to the end of the cycle, at point F. It is during this phase that the melt solidifies gradually, as the coolant that circulates within the cooling channels in the mold removes the heat. The cooling and solidification rates determine the decreasing speed of the cavity pressure.

The increased demand for micro-scale parts and devices is being met, in many cases, by the micro-injection molding of polymer parts [7,8,9]. However, part inspection is difficult, due to the micro-scale dimension in the micro-injection molding process. In addition, process control also becomes challenging, since the process is susceptible to slight changes in the process parameters, such as the mold temperature, injection velocity, and packing pressure [10,11,12,13]. 

At present, a suitable process monitoring method, such as cavity pressure monitoring, can be employed to detect any process deviation that may cause defects in the part quality. In the process of injection molding, the constant process parameters cannot guarantee the consistency of product quality, but the process parameters changed by cavity pressure can greatly improve the quality of the products. The cavity pressure in the injection molding process is the result of the combined action of various process parameters. From the cavity pressure curve, we can directly see the injection process, the switching point of injection to packing pressure, the packing pressure process, and the cooling process, and it finally becomes a comprehensive index to measure the quality of the products [14,15]. Through the change in the pressure curve, the quality of the parts can be monitored, which can be used to analyze the quality, size, flash, shrinkage, shot, and warpage of the parts, providing the most stable process parameters for the cavity. The main advantages of using cavity pressure technology are as follows: it can reduce the times of mold testing and improve the efficiency of mold testing. It can realize rapid process reproduction under different machines and working environments. It can ensure the production of high-quality and high-qualified rate products.

Cavity pressure has been found to be a reliable process indicator in injection molding, for both part quality and process monitoring [16,17,18,19,20,21,22,23,24,25,26,27]. Specifically, it has been found to provide real-time detection of part and process deviation. As such, the cavity pressure measurement holds potential for monitoring the part quality in micro-injection molding, where direct part inspection is difficult and often costly, due to part handling issues and microscopic feature sizes. Huang [14] presented a simple grey model to predict, instantaneously, the volumetric-filling point when monitoring the cavity pressure profile in each molding. It was found to be a good indicator of product quality; the cavity pressure profile is applied here, to obtain a more precise switchover control. Kurt. et al. [15] studied the relationship between cavity pressure and shrinkage, by using the cavity pressure curve. The results indicate that cavity pressure and mold temperature are the dominant factors determining the quality of the final product. Kim. et al. [16] showed the influences of injection flow rate, peak cavity pressure, melt temperature, and mold temperature on the filling length. The cavity pressure and the temperature transition of the melt in the micro channels had a critical influence on the filling process. Hassan [17] investigated the effect of injection molding parameters on the polymer pressure inside the mold cavity. The results indicate that the cavity pressure and product weight increase with an increase in the packing pressure, packing time, and injection pressure, for all the analyzed polymers. Chen et al. [18] proposes a quality index for online quality monitoring and prediction purposes, based on the pressure, viscosity, and energy features extracted from the pressure profiles obtained at the load cell, nozzle, and molding cavity, respectively. It is shown that all of the quality indexes are correlated with the injection-molded quality, and hence provide a feasible basis for the realization of an on-line quality monitoring and control system. Injection molding part quality is modeled using a multivariate sensor by Gordon et al. [19]. The analysis indicates that the most important process data are gathered from in-mold sensors, where the acquired information is closest to the states of the polymer molding the final product. Wang et al. [20] investigated the influence of mold temperature, melt temperature, packing pressure, packing time and peak cavity pressure, on the weight of micro-injection molding products, by the Taguchi orthogonal experiment. The experimental results show that the packing pressure increased from 85 MPa to 100 MPa, the weight of the tensile specimen increased from 0.544 g to 0.559 g, increased by 2.7%, and the weight of the impact specimen increased from 0.418 g to 0.425 g, increased by 1.7%. Zhao et al. [21] proposes a non-destructive method for measuring cavity pressure, by evaluating the stress on the tie bars of the injection molding machine, using ultrasonic technology. The direct monitoring of process signals during micro-injection was addressed via pressure and temperature sensors that were placed in the following two different mold locations: the runner system and the mold micro-featured cavity by Mendibil et al. [22]. It has been observed that both the runner system and micro-featured cavity pressure signals are linked to the replication quality level of the micro-injected part, and show similar performance, in terms of part quality differentiation. The process parameter that causes the greatest variations is the temperature set-point of the machine nozzle. Gao et al. [23] demonstrates an in-process sensing technique for online product quality assessment. The system measures four parameters within the injection mold cavity, which are directly correlated with the part quality, and they are as follows: melt pressure, temperature, velocity, and viscosity. Gim et al. [24] tried to detect the filling imbalance by temperature sensors in the runner, and indirect pressure sensors at the bottom of the lens core. The temperature signal showed a reliable correlation with respect to the resin-arrival-time difference, but the pressure signal did not produce a reliable result. Guan et al. [25] proposed that the mold surface strain profile could indicate the part weight or thickness and the critical time when the part surface lost contact with the cavity surface in a large area. The monitoring of the mold surface strain could serve as an interesting alternative to the direct monitoring of the cavity pressure, with respect to process and part quality control for ICM.

As the characteristic value of cavity pressure, the peak cavity pressure and area under the curve can be used to detect the qualified range of products in the paper. The linear relationship between various process parameters and the cavity pressure was explored, and accurately adjusted the process parameters, by calculating the integral value, so as to greatly improve the qualified rate of the products.

## 2. Experiments

### 2.1. Materials

The sample material used in this work was polypropylene in the form of pellets and with a trade mark 5090T (MFI = 15 g/10 min), supplied by the Formosa petrochemical Corp, Taiwan, China.

### 2.2. Micro-Injection Moulding Experiments

Micro-injection molding machine: the experimental work was carried out on an injection molding machine of type BOY XS, Germany concept having a maximum injection pressure 2298 bar, with screw diameter for plastication 14 mm and maximum weight of the product 6.1 g as shown in Figure 2. 

Mold: The multi-spline injection mold constructed from two parts (tensile specimen and impact specimen). The mold cavity thickness is 1 mm. The cavity pressure and temperature are measured in the mold cavity by the quartz sensor for mold cavity pressure type Kistler 6190CA, which has a front of 4.0 mm diameter. Data output from the amplifier is collected using a Kistler 5865 Como injection system. Computer is used to record the output reading of the acquisition system through an interface cart by the help of lab view program. The dimensions of specimen and gate are shown in Figure 3. 

Mold temperature controller: The mold temperature controller (model TP6ZE) was adopted using PIOVAN Co. Ltd, Italia.

Chiller (model ML-CA03) was adopted Ming Lee Co. Ltd, Hong Kong, China.

Electronic balance (model CP214) was adopted OHAUS Co. Ltd, America. The accuracy is 0.1 mg.

### 2.3. Parameters Setting

Melt temperature, mold temperature, packing pressure were chosen as the process parameters. Parameters of injection molding process are reported in Table 2.

Orthogonal design is a high-efficiency design method for tests to arrange multi-factor tests and seek optimal level combinations. The design method of orthogonal tests is able to determine optimal parameters by simply calculating influences of each factor on test results, showing the influences in charts, and then comprehensively comparing differences. The calculation is carried out on orthogonal tables, so the whole process is simple and clear. In this way, enough information can be obtained through a few tests, thus saving costs. An L_16_(4^3^) orthogonal array was selected for the experimental design for each of the three factors. The four levels for the three parameters were identified during the 16 experiments. The values of these parameters are presented in Table 3. The twenty samples were collected for each run, after the machine had been allowed to reach steady state.

## 3. Results and Discussions

In the process of injection molding, the pressure curves of different molds are not the same, but the general trend of the pressure curves of different molds is the same. They can reflect the flow of melt in the mold cavity and the state of each injection molding shot. It can be observed from the figure that injection molding is a dynamic cycle process, from injection to pressure holding and then to cooling. In addition to observing the general shape of the curve, cavity pressure data are correlated to part quality by deriving quantitative values representing the features of the curve. In the present study, the peak pressure and area under the curve are the two attributes obtained from the curve, as shown in Figure 4.

### 3.1. Effect of Different Injection Velocity

Figure 5 shows the results obtained from two different injection velocity trials. The packing pressure is 60 MPa, the melt temperature is 210 °C, and the mold temperature is 40 °C. The injection velocity is 35% and 45%, respectively. The 35% and 45% injection velocities represent thirty-five percent and forty-five percent of the machine maximum injection rate. All the molding parameters were unchanged, except injection velocity. As shown in the figure, the trial with the higher injection velocity reaches the packing phase and cooling phase earlier. Because this is higher, the injection rate injects the material faster and causes the cavity to be filled faster. The difference in filling rates between the trials is easily noticeable from the figure. The red line (higher injection velocity) has a steeper slope at the filling stage. Despite the difference in the filling stage, both the trials share the same cooling rate (slope of cooling phase) and final cavity pressure value.

### 3.2. Effect of Different Pack Pressure

Figure 6 shows how pressure curves respond to different pack pressure settings. Here, only the pack pressure settings were varied, while the other settings remained unchanged. As can be observed from the figure, the three curves are similar before the switchover point. The curves start to behave differently after the switchover happens, due to different pack pressure settings. After the switchover point, the entire filling phase is solely relying on pack pressure. As a result, a higher pack pressure produces a higher cavity pressure and faster filling rate. The result is reasonable, as the higher pack pressure forces more polymer melt into the cavity in a shorter time frame. The 90 MPa line reaches the packing stage the earliest, as the filling is completed by the higher pack pressure. The high pack pressure also produces a higher peak cavity pressure value. The 60 MPa line reaches the packing phase slower and has a lower peak cavity pressure. Here, the pack pressure is lower, it is still able to fill up the part completely, at a slower rate. On the other hand, the 0 MPa line does not reach the packing phase at all. From Figure 6, it is clear that the polymer melt stops filling the cavity once the switchover takes place, causing the pressure to drop immediately. It can be deduced, from the cavity pressure curve behavior, that a short part is produced from this trial.

### 3.3. Effect of Different Switchover Setting 

The following discussion focuses on the relationship between peak cavity pressure and switchover settings. The peak cavity pressure increases when the switchover occurs later in the cycle. This result corresponds with the initial prediction that a later switchover allows the polymer melt to fill up more of the cavity, and hence generate a higher peak cavity pressure. The pressure curves with a switchover of 4 mm have a relatively low peak pressure value and a small area under the curve, in Figure 7. This reflects the fact that very little polymer melt was able to fill up the cavity. For pressure curves with a later switchover of 10 mm, a sharp rise in pressure is observed. This indicates that the polymer melt has at least filled up the thin wall section. Given that there is no pack pressure during the molding process, the polymer melt stops filling once the switchover takes place. As a result, a later switchover allows the injection to proceed longer and allows the melt to flow further into the cavity before the injection phase ends.

In terms of part quality, a normal visual inspection can also easily detect some obvious defects on the parts. In the cavity pressure curve corresponding to short-shot 3, the cavity pressure peak is too small, and the molten polymer cannot fill the whole cavity, that is, the experimental spline of short-shot 3 has a very low cavity pressure peak. For the experimental splines of short-shot 1 and short-shot 2, the filling stage was completed and the pressure holding stage was entered. The slope of the pressure curve and the pressure value of the mold cavity increased at the same time, but the pressure holding stage was incomplete, the end and detail parts of the product were not filled completely, and the product density was low. The pressure peak value of the cavity corresponding to the short-shot 1 and short-shot 2 fails to reach the peak value in the case of complete filling, and the increased degree of the slope of the pressure curve is insufficient.

### 3.4. Relationship between Cavity Pressure and Part Weight 

Two attributes obtained from the cavity pressure—peak cavity pressure and area under curve—are utilized to relate with the part weight. Figure 8 shows the average part weights and average value of the area under the curve for all the trials, while Figure 9 shows the average weights and average peak cavity pressure. Both the figures show that both the attributes appear to have promising correlation with part weight, as shown in the figures, respectively; the higher part weight has the higher value in both the attributes. In terms of the standard error of every trial for average part weight, average peak cavity pressure, and average area under the curve, the error values are not presented in Figure 8 and Figure 9, because they are relatively small and hardly noticeable from the plots. The small value of standard error signifies that the average value obtained has small scattering and deviation from the mean value.

To further investigate the relationship between cavity pressure and part weight, the coefficient of determination (R^2^ value) was calculated, to determine how well the peak cavity pressure and area under the curve correlate with the part weight. The coefficient of determination is a measure of the degree of correlation or dependence between the dependent and independent variables in a regression analysis. A high R^2^ value indicates that the two variables are well correlated. The higher R^2^ value shows that a later switchover in the injection molding process not only produces a better quality part, but also produces a better correlation between part quality and cavity pressure (both peak value and area under the curve). A plot to show the correlation of the average area under the curve and the peak cavity pressure, with respect to part weight, is presented in Figure 10. The distribution of all the shots of both attributes in the DOE trial with respect to part weight. As shown in the figure, both the attributes have good correlation with the part weight. 

Under the current processing parameter range, both the attributes are found to have a second-order polynomial relationship with part weight. The curves start to increase linearly at the beginning and then begin to flatten out. The flat part corresponds to trail 5, trail 7, trail 10, and trail 13. In the four experimental parameter settings, the set values of melt temperature and packing pressure are higher, that is, a higher melt temperature and packing pressure can make more molten polymer fill the cavity in the lower response pressure range. Although both attributes respond to part weight in an almost similar manner, the R^2^ values are different. The area under the curve is found to have a higher R^2^ value of 0.856 compared to the peak cavity pressure, with an R^2^ value of 0.77. This signifies that area is a better quality characterization than peak cavity pressure. 

A possible explanation to this difference is the significant effect of pack pressure. Pack pressure provides extra material to compensate for the part shrinkage in the cavity. However, this action is not well represented by the peak cavity pressure, as it only indicates the maximum pressure at the moment the part is fully filled; anything that occurs after the part is filled is overlooked. On the other hand, the area under curve covers the entire processing window, from the moment the melt enters the cavity to the time when the part solidifies. In the current trials, pack pressure appears to have an important effect on the part weight, therefore the area appears to be a better attribute to correlate to part quality.

Here, only the area under curve will be presented, since it has a higher R^2^ value, as discussed earlier. In general, the result obtained from Figure 11 is expected, since trials with low level settings, such as trial 1, trial 2, trial 6, trial 11, and trial 16, produced lighter parts and lower attribute values, while trials with higher level settings, such as trial 5, trial 7, trial 10, and trial 13, produced heavier parts and a greater area under the curve. As shown in the figure, two distinct clusters are observed from in the data, with most of the parts weighing from ~530 mg to ~580 mg and a second group falling in the range from ~590 mg to ~630 mg. This second group consists of trial 3, trial 7, trial 10, and trial 13. Both of these trials were conducted at high-level settings in two of the three processing parameters. The trail 5 has a large area under the curve, but a lower weight, which may be due to the greater influence of the packing pressure. The separation of the trials suggests the possibility of having an excessive gap between each level for the pack pressure setting. Despite the disjointed data, the results show that pack pressure has the greatest effect on part weight.

### 3.5. Relationship between Part Weight and Runner Weight

Figure 12 presents the relationship between part weight and runner weight. As shown, part weight and runner weight have little to no relationship between them. This means that the weight of the product is not affected by the weight of the runner system. A logical explanation to this dissimilarity is the pack pressure setting. When pack pressure is involved in the injection molding process, at the switchover point, the screw will continue to move forward for a certain distance and speed, to attain the required pack pressure. 

## 4. Conclusions

An experimental work is carried out to study the quality characterization method of micro-injection products based on cavity pressure in this paper. In terms of the relationship between cavity pressure and part weight, the peak cavity pressure and area under the curve were the two attributes that were used to correlate to part quality. The following items summarize the major finding of this experiment:(1)A later switchover allowed the injection to proceed longer and produce heavier parts;(2)By comparing different cavity pressure curves, the general shapes of the curves were able to indicate different types of shortages produced;(3)The coefficient of determination of part weight, for peak cavity pressure and area under the curve, were 0.77 and 0.856, respectively. This showed that the area under the curve appeared to be a better process and quality indicator than peak cavity pressure.

## Figures and Tables

**Figure 1 polymers-13-02755-f001:**
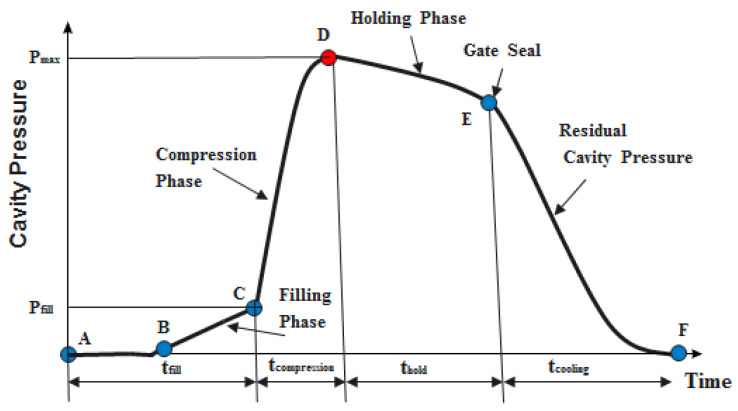
The typical cavity pressure profile, see A–F in Table 1.

**Figure 2 polymers-13-02755-f002:**
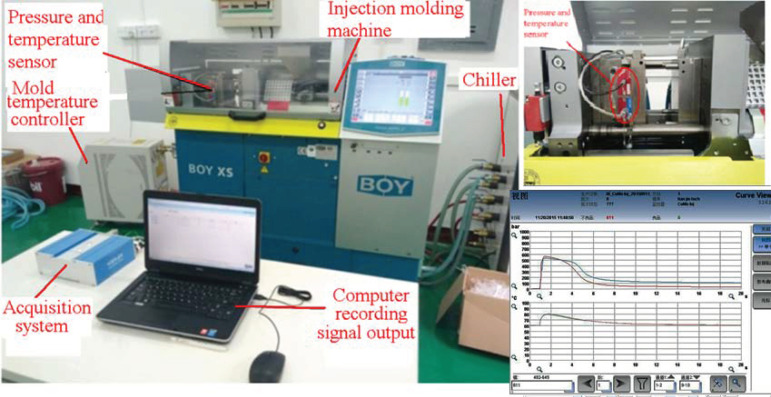
Experimental setup.

**Figure 3 polymers-13-02755-f003:**
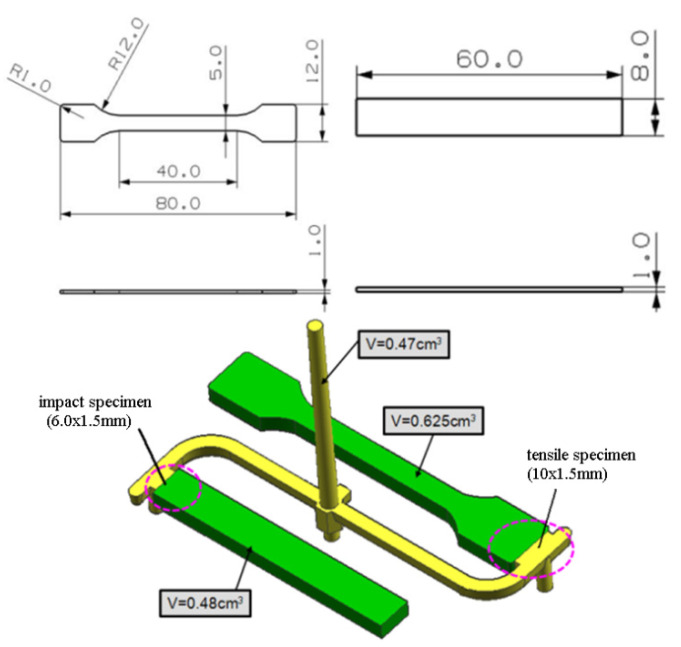
Dimension of specimen and gate.

**Figure 4 polymers-13-02755-f004:**
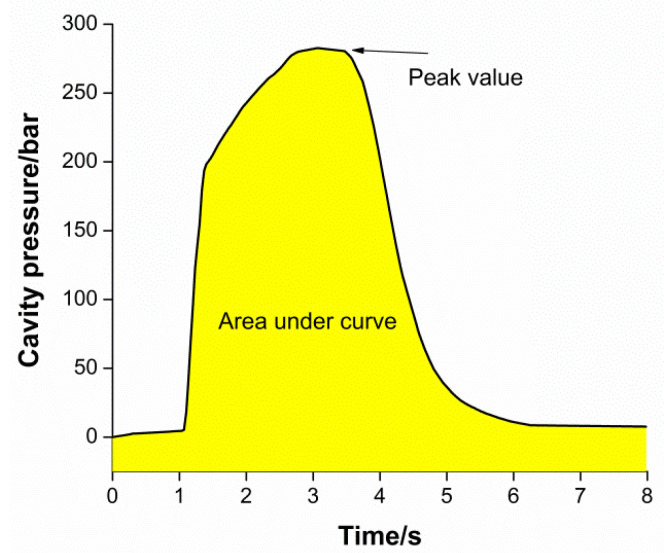
Graphical representation of peak pressure and area under curve.

**Figure 5 polymers-13-02755-f005:**
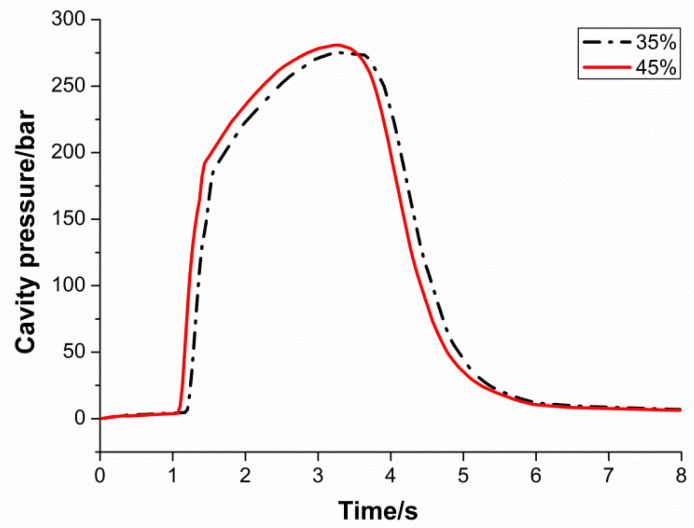
Influence of injection velocity on cavity pressure.

**Figure 6 polymers-13-02755-f006:**
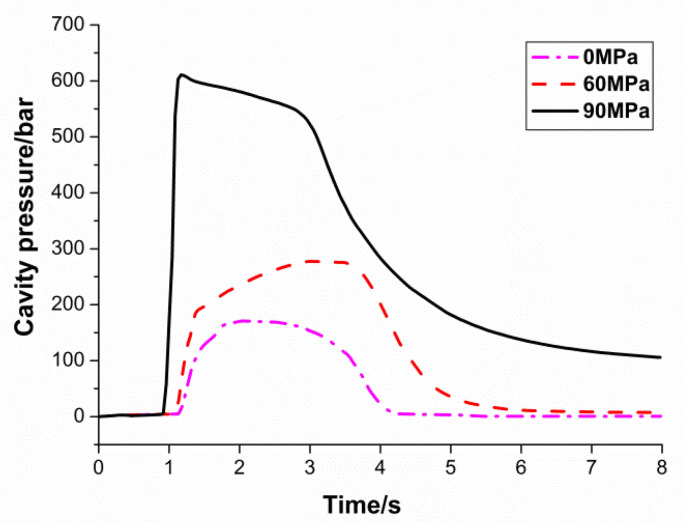
Influence of pack pressure on cavity pressure.

**Figure 7 polymers-13-02755-f007:**
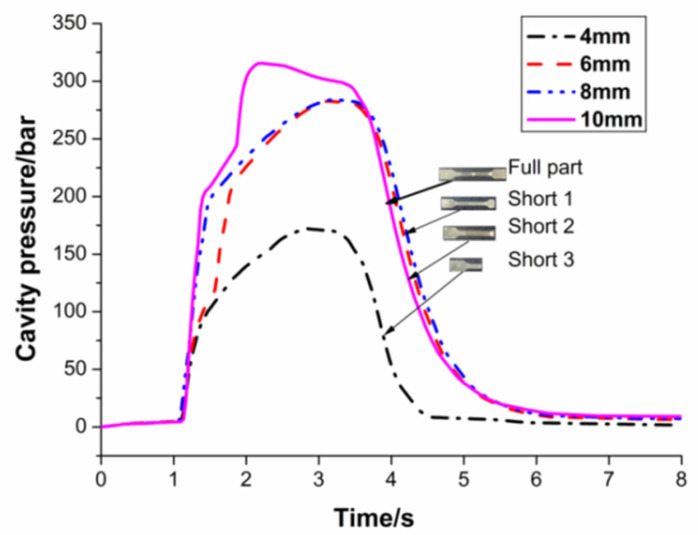
Influence of switchover setting on cavity pressure.

**Figure 8 polymers-13-02755-f008:**
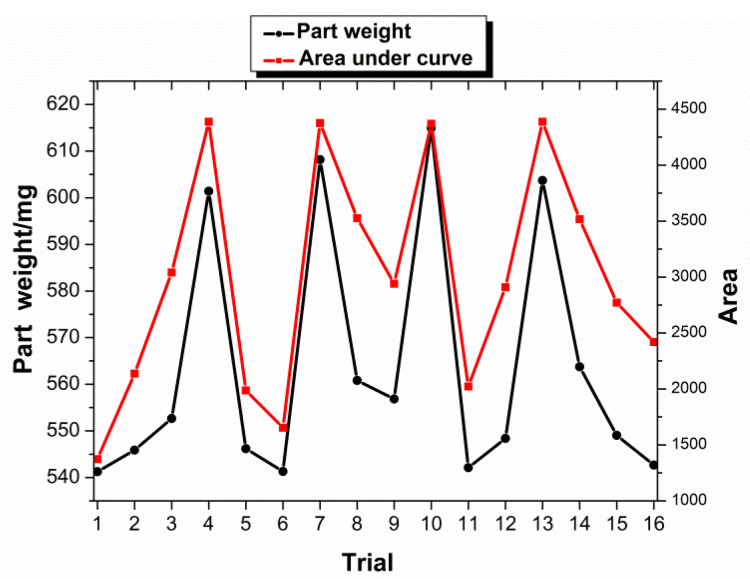
Area under curve and part weight for the trials.

**Figure 9 polymers-13-02755-f009:**
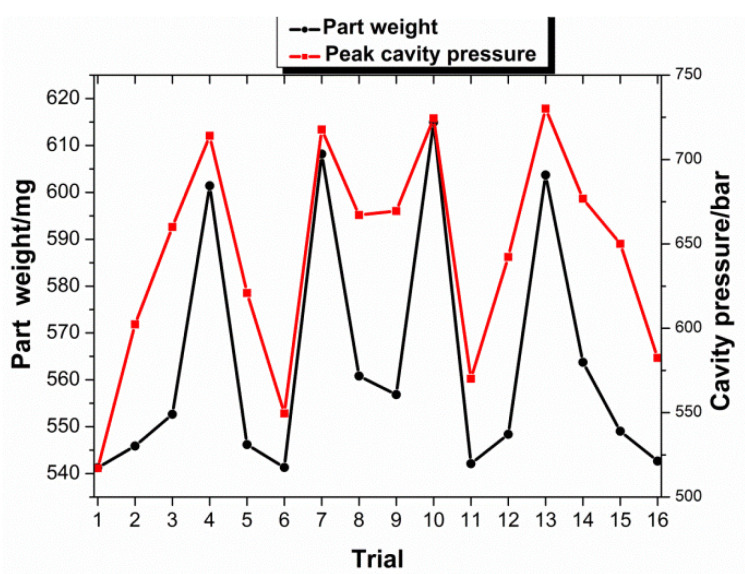
Pear cavity pressure and part weight for the trials.

**Figure 10 polymers-13-02755-f010:**
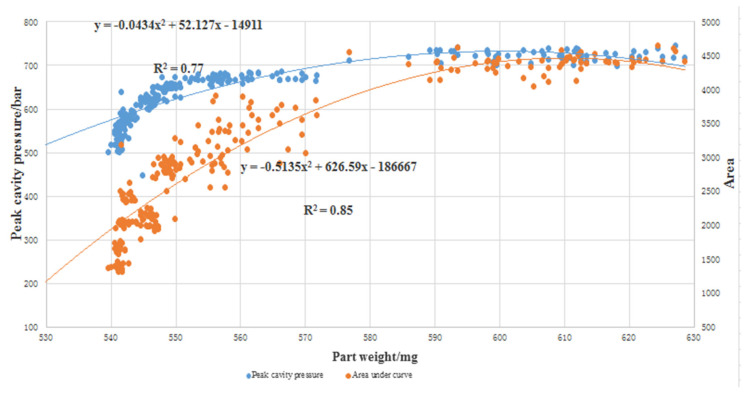
Area under curve and peak cavity pressure vs. part weight.

**Figure 11 polymers-13-02755-f011:**
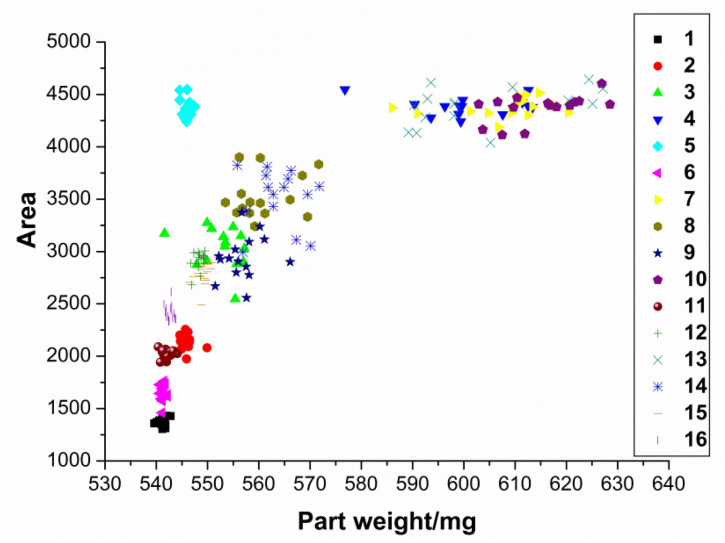
Area under curve vs. part weight of DOE trial.

**Figure 12 polymers-13-02755-f012:**
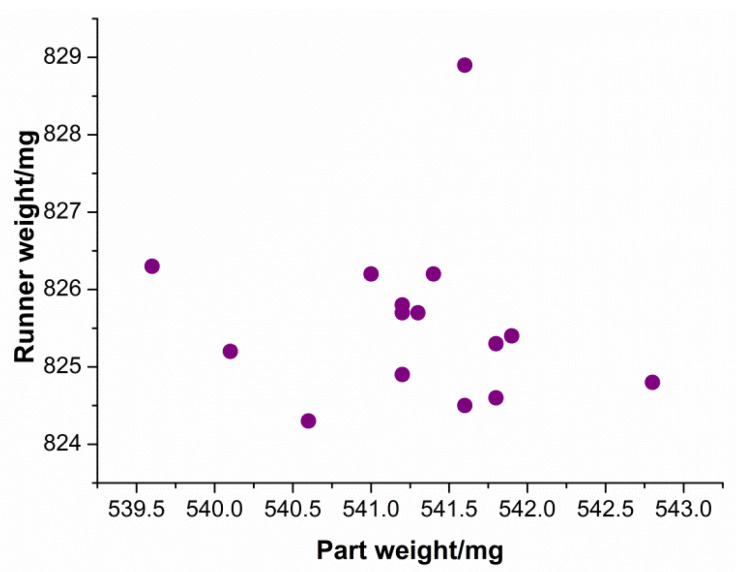
Runner weight vs. part weight.

**Table 1 polymers-13-02755-t001:** Cyclic process of injection molding.

Point	Explain
A	The filling process starts
B	The cavity pressure signals begin
C	The filling phase is complete
D	Peak cavity pressure value
E	Gate frozen
F	Molding end

**Table 2 polymers-13-02755-t002:** Parameters of injection molding process.

Run	Melt Temperature(°C)	Mold Temperature(°C)	Packing Pressure(MPa)
1	210	30	80
2	220	40	90
3	230	50	100
4	240	60	110

**Table 3 polymers-13-02755-t003:** The micro-injection molding parameters and corresponding levels.

Run	Melt Temperature (°C)	Mold Temperature (°C)	Packing Pressure (MPa)
1	210	30	80
2	210	40	90
3	210	50	100
4	210	60	110
5	220	30	90
6	220	40	80
7	220	50	110
8	220	60	100
9	230	30	100
10	230	40	110
11	230	50	80
12	230	60	90
13	240	30	110
14	240	40	100
15	240	50	90
16	240	60	80

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
