# Peer review of "Research on Quality Characterization Method of Micro-Injection Products Based on Cavity Pressure"

_polymers, 2021, doi:10.3390/polym13162755_

Round 1

Reviewer 1 Report

In the newly submitted version, the paper quality has been improved and can be considered for publication. Before that, please note:

1) There are still some minor English typos/ grammatical mistakes. Please proofread the manuscript carefully.
2) Some figures, e.g., Fig. 3 and Fig. 10, do not have acceptable readability/quality. Please improve their quality.

Author Response

In the newly submitted version, the paper quality has been improved and can be considered for publication. Before that, please note:

1) There are still some minor English typos/ grammatical mistakes. Please proofread the manuscript carefully.

Answer:I have revised.

Line 136, “process reported in Table 2.” revised “process are reported in Table 2.”

Line 151: “each injection molding process.” revised” each injection molding shot.”

Line 246 missed spacing.  “trial 11and “ revised “trial 11 and”

2) Some figures, e.g., Fig. 3 and Fig. 10, do not have acceptable readability/quality. Please improve their quality.

Answer:I have improved figures quality.

Thank you very much.

Reviewer 2 Report

The present paper deals of the application of cavity pressure as indicator of injection molded parts quality. The argument is interesting but, from the paper, it is not evident the novelty introduced from authors. It seems like an application of known methods.  Could you clarify this aspect?

The meaning of some sentences is not clear as lines 13-15, 108-111, 161-163.

In lines 18-20, I think that the peak cavity pressure and are under pressure curve are in correlation with part weight and not with cavity pressure as reported.  

The last keyword, “area under curve”, is not precise and it is too vague.

I not agree very much with the injection process phase reported from authors that are “filling, melt pressing (or packing), holding, and cooling (lines 35-36). Usually the four phases are defined such as filling, maintaining pressure, cooling and demoulding. The “compression phase”, considered by authors, could introduce some misunderstanding with holding pressure and it is usually considered inside the filling phase. So, is the packing pressure, on which the results are based, measured during filling phase? In the packing pressure the injection pressure? Why did you choose this notation?

Before or after reference numbers (14-15-16…) the space is often missed.

I suggest to add the physical properties of the used polymer as melt temperature, MVR, density.

Please add the number of the International standard followed to design the specimens.

Fig. 3 is not clear and, in particular, are reported the dimensions of the gates at start of both specimens?

Line 136, miss “are”.

How were selected the process parameters range? Did you perform screening tests?

Line 151: I suggest “shot” instead of “process”.

Line 160: how was the injection velocity defined? 35% and 45% of which? are they derived from pressure measurements?

Could you show and add the pictures of the parts realized in the 3 conditions reported in figure 6? While the pictures of Fig. 7 need to be enlarged.

In lines 210 and following you report that the standard errors are too small to be reported in the graph. I suggest to add a table with these values.

Were the part weights measured together for both samples (tensile and impact)? And was the runner system removed?

Lines 236-237. The author report: “A possible explanation to this difference is the significant effect of pack pressure. Pack pressure provides extra material to compensate for the part shrinkage in the cavity.”  Thus, is the pack pressure evaluated during holding phase? In that phase the shrinkage is compensated not during filling.

The aim of Section 3.5 is not clear. There are no reported results.

Author Response

The present paper deals of the application of cavity pressure as indicator of injection molded parts quality. The argument is interesting but, from the paper, it is not evident the novelty introduced from authors. It seems like an application of known methods.  Could you clarify this aspect?

Answer:Although there is this method, there are few reports on the specific research content.

The meaning of some sentences is not clear as lines 13-15, 108-111, 161-163.

Answer:I have revised.

In lines 18-20, I think that the peak cavity pressure and are under pressure curve are in correlation with part weight and not with cavity pressure as reported.  

Answer:the peak cavity pressure and are under pressure curve are the two attributes of cavity pressure.

The last keyword, “area under curve”, is not precise and it is too vague.

Answer:I have revised.

I not agree very much with the injection process phase reported from authors that are “filling, melt pressing (or packing), holding, and cooling (lines 35-36). Usually the four phases are defined such as filling, maintaining pressure, cooling and demoulding. The “compression phase”, considered by authors, could introduce some misunderstanding with holding pressure and it is usually considered inside the filling phase. So, is the packing pressure, on which the results are based, measured during filling phase? In the packing pressure the injection pressure? Why did you choose this notation?

Answer:The injection molding process includes the above four aspects, which may only express the direct difference.

Before or after reference numbers (14-15-16…) the space is often missed.

Answer:I have revised.

I suggest to add the physical properties of the used polymer as melt temperature, MVR, density.

Please add the number of the International standard followed to design the specimens.

Answer:number of  the specimens  are two.

Fig. 3 is not clear and, in particular, are reported the dimensions of the gates at start of both specimens?

Answer: The dimensions of the gates have been marked. 6.0x1.5mm and 10x1.5mm.

Line 136, miss “are”.

Answer:I have revised.

How were selected the process parameters range? Did you perform screening tests?

Answer:I have performed screening tests.

Line 151: I suggest “shot” instead of “process”.

Answer:I have revised.

Line 160: how was the injection velocity defined? 35% and 45% of which? are they derived from pressure measurements?

Answer:The 35% and 45% of injection velocity represent thirty five percent and forty five percent of the machine maximum injection rate.

Could you show and add the pictures of the parts realized in the 3 conditions reported in figure 6? While the pictures of Fig. 7 need to be enlarged.

Answer:Because the product is very small, the picture of the product is similar to figure 7.

In lines 210 and following you report that the standard errors are too small to be reported in the graph. I suggest to add a table with these values.

Were the part weights measured together for both samples (tensile and impact)? And was the runner system removed?

Answer:The weight of the product is measured separately. runner system are removed.

Lines 236-237. The author report: “A possible explanation to this difference is the significant effect of pack pressure. Pack pressure provides extra material to compensate for the part shrinkage in the cavity.”  Thus, is the pack pressure evaluated during holding phase? In that phase the shrinkage is compensated not during filling.

The aim of Section 3.5 is not clear. There are no reported results.

Thank you very much.

Reviewer 3 Report

Wang et al put forward the cavity pressure profile as a measure for checking the success of the micro-injection. They focus on the weight. As such the work is a bit short but has merit in view of finding correlation.

L 31 please add Polymers 2019, 11, 87 and Polymers 2021, 13, 1541; also Plast. Rubber. Comp. 2021 50, 137 and Polymers 2021, 13, 398;

For a general reader please add a figure of an injection unit aside so that link with P variation can be made more clearly

I guess this a revised version. I was not involved in the first review process (I see highlighted changes).

What is the error or Figure 4?

General comment: would be good to mention Table 3 are targets or average parameters.

“The injection velocity is 35% and 45%” For a general reader you need to say with respect to what.

Figure 10: make 2 superscript twice

Figure 10 needs a bit more discussion: R^2 value as such (I mean vs. e.g. 0.95 “standard” value) as shape of line, I mean physical or not (especially at the end).

L 246 check spacing

Figure 12 has significant scatter. Please comment more on this.

Author Response

Wang et al put forward the cavity pressure profile as a measure for checking the success of the micro-injection. They focus on the weight. As such the work is a bit short but has merit in view of finding correlation.

L 31 please add Polymers 2019, 11, 87 and Polymers 2021, 13, 1541; also Plast. Rubber. Comp. 2021 50, 137 and Polymers 2021, 13, 398;

Answer:Please provide the name of the paper, thank you.

For a general reader please add a figure of an injection unit aside so that link with P variation can be made more clearly

I guess this a revised version. I was not involved in the first review process (I see highlighted changes).

Answer:Yes, it is a revised version.

What is the error or Figure 4? 
Answer:Figure 4 is no error.

General comment: would be good to mention Table 3 are targets or average parameters.

Answer:I have revised.

“The injection velocity is 35% and 45%” For a general reader you need to say with respect to what.

Answer:35% and 45% represent thirty five percent and forty five percent of the machine maximum injection rate.

Figure 10: make 2 superscript twice

Answer:I have revised in figure 10.

Figure 10 needs a bit more discussion: R^2 value as such (I mean vs. e.g. 0.95 “standard” value) as shape of line, I mean physical or not (especially at the end).

Answer:it is no physical.

L 246 check spacing

Answer:I have revised.

Figure 12 has significant scatter. Please comment more on this.

Answer:I have revised.

Thank you very much.

This manuscript is a resubmission of an earlier submission. The following is a list of the peer review reports and author responses from that submission.

Round 1

